# Investigation of the Thermal Properties of Diesters from Methanol, 1-Pentanol, and 1-Decanol as Sustainable Phase Change Materials

**DOI:** 10.3390/ma13040810

**Published:** 2020-02-11

**Authors:** Rebecca Ravotti, Oliver Fellmann, Ludger J. Fischer, Jörg Worlitschek, Anastasia Stamatiou

**Affiliations:** Competence Centre Thermal Energy Storage (TES), Lucerne University of Applied Sciences and Arts, 6048 Horw, Switzerland; oliver.fellmann@hslu.ch (O.F.); ludger.fischer@hslu.ch (L.J.F.); joerg.worlitschek@hslu.ch (J.W.); anastasia.stamatiou@hslu.ch (A.S.)

**Keywords:** phase change materials, latent heat storage, thermal energy storage, sustainability, energy, esters, diesters, PCM, LHS, TES

## Abstract

Esters present several advantages when compared to traditional materials used for thermal energy storage, amongst which are better sustainability and greater chemical stability. However, at present, their thermal properties remain mostly unknown or not well documented. In this study, 12 diesters from four dicarboxylic acids (oxalic, succinic, suberic, sebacic) and three alcohols (methanol, 1-pentanol, 1-decanol) have been assessed as bio-based phase change materials for thermal energy storage. All diesters have been synthesized via Fischer esterification to high purities, and their chemical structures, as well as thermal properties, have been fully characterized. The diesters investigated show phase change transitions in a low–mid temperature range between −32 and 46 °C with maximum enthalpies of 172 J/g and show higher degrees of supercooling compared to fatty monoesters. Similarly to other esters classes, some trends correlating the chemical structures to the thermal properties were identified, which would allow for the development of property prediction tools.

## 1. Introduction

In view of the 2020 and 2050 European energy goals set by policymakers to allow for a more sustainable future less dependent on fossil fuels, energy storage technologies will play a greater role in the upcoming years [1,2,3,4,5,6,7,8]. In particular, thermal energy storage (TES) is of key importance in covering for the mismatch between demand and supply, especially in the case of renewable energy sources [9]. Amongst other types of TES, latent heat storage (LHS) with phase change materials (PCM) presents the advantage of allowing achieving more compact solutions. This is of critical importance in the context of urban realities where the available space is limited [10].

To improve the efficiency of LHS, the appropriate PCMs have to be selected. Some of the desirable properties are thermal stability, narrow phase change transitions, low supercooling, high heat of fusion, chemical stability, non-toxicity and preferably bio-origin or recyclability [11,12]. While salt hydrates and paraffins are well-established PCMs for LHS applications, they present some disadvantages such as segregation, supercooling, and flammability [13,14]. Therefore, the possibility of alternative PCMs is currently being evaluated.

Amongst organic materials, carboxylic esters are a class of bio-based compounds naturally occurring in oils and waxes. They are derived from carboxylic acids and alcohols, but unlike both acids and alcohols, they present the advantage of showing lower reactivity, thus being less prone to corrosion and oxidation. As a result of these attractive properties and virtually millions of possible combinations between alcohols and acids to generate esters, researchers have turned their attention to this class of compounds as a possible alternative to traditional PCMs.

To date, most researchers have focused their studies on characterizing the thermal properties of esters from fatty carboxylic acids due to their commercial availability and the possibility to extract them from renewable feedstock. Feldman et al. [15] investigated 12 esters from mixtures of stearic and palmitic acid with methanol, propanol, and butanol. They observed a high thermal stability with low supercooling rates and enthalpies of fusion of around 150 J/g in a low–mid temperature range between 20 and 32 °C. Sari et al. [16] synthesized and studied esters from stearic acid with isopropanol, n-butanol, and glycerol, and obtained similar results. Other studies include those of Stamatiou et al. [17], where the authors investigated 11 commercial saturated unbranched esters, and Ravotti et al. [18,19], where 15 carboxylic fatty esters were synthesized and their thermal properties were characterized. In general, these authors measured enthalpies of fusion between 150 and 190 J/g in a temperature range of 10–50 °C, which would be suitable for low–mid temperature applications such as space heating.

However, there are other types of esters, and linear carboxylic fatty esters represent only a small percentage of this group of compounds. In this regard, only few studies have been conducted, and the data available on other classes of esters is limited. Ravotti et al. [20] synthesized and assessed the thermal behavior of six lactones (cyclic esters). Lactones showed a general tendency to degrade in the same temperature range as their phase change, which renders them unsuitable for LHS applications. Suppes et al. [21] evaluated the suitability of triglycerides from stearic and palmitic acids, and they compared their performance with fatty monoesters as well. Triglycerides have latent heats of fusion comparable to those of paraffins, but they are prone to polymorphism. Yet data reported in previous studies from Malkin et al. [22] and Ferguson et al. [23] suggest that the polymorphism of triglycerides can be controlled and the thermodynamically stable phases can be triggered. Finally, Aydin et al. [24] synthesized and characterized the thermal properties of high-chain diesters from tetradecanol and observed melting temperatures between 50 and 58 °C and enthalpies of fusion above 200 kJ/kg.

Despite recent advances, data on the thermal properties of other ester classes beside fatty esters is still limited or completely lacking. Therefore, it is of the utmost importance to acquire the missing thermal data in order to obtain a more comprehensive overview of esters as possible PCM for LHS applications.

In this study, the thermal properties of 12 diesters synthesized from oxalic acid (C_2_ = carbon chain length equal to two), succinic acid (C_4_), suberic acid (C_8_), and sebacic acid (C_10_) each coupled with three alcohols of different chain lengths, namely methanol (C_1_), 1-pentanol (C_5_), and 1-decanol (C_10_) are studied. The names are abbreviated as follows: Di + alcohol abbreviation + acid abbreviation. For example, MeOH indicates methanol and Ox indicates oxalic acid; therefore, DiMeOHOx stands for dimethyl oxalate. In particular, MeOH = methanol, PeOH = 1-Pentanol, DeOH = 1-Decanol, Ox = oxalic acid, Su = succinic acid, Sub = suberic acid, Se = sebacic acid (see chemical structures and IUPAC names in Appendix A
Figure A1). The alcohols of provenance have been chosen in order to allow the diesters to be comparable to the fatty esters synthesized by Ravotti et al. [18,19]. The dicarboxylic acids of origin were selected based on their natural occurrence with increasing carbon number in order to evaluate the influence of the aliphatic chains’ lengths on the thermal properties of the diesters produced.

All the diesters have been synthesized via Fischer esterification, and their purities and chemical structures have been confirmed via Attenuated Total Reflectance Infrared Spectroscopy (ATR-IR) and Gas Chromatography coupled with Mass Spectrometry (GC-MS). The thermal properties have been assessed via Differential Scanning Calorimetry (DSC) and Thermogravimetric Analysis (TGA).

To the best of the authors’ knowledge, this is the first time that the thermal properties of the compounds mentioned above are investigated, with the exception of dimethyl sebacate (DiMeOHSe) [25].

## 2. Results and Discussion

### 2.1. Structural Characterization

All diesters presented were synthesized in house according to the synthesis proposed by Ravotti et al. [18,19] for saturated fatty esters. Given that the synthesis procedure had been optimized for fatty esters and not diesters, its success in the case of diesters had to be validated first. To do so, DiMeOHOx, a commercially available diester, was both purchased and synthesized in house, and its properties compared in terms of chemical structure and thermal behavior. The synthesis and purification procedures were considered valid when exact matches between the Infrared (IR) and Mass spectrometry (MS) spectra of the commercial and synthesized DiMeOHOx were obtained, thus confirming the chemical structure of the compound produced.

#### 2.1.1. ATR-IR

To confirm the formation of the ester bond and the absence of unreacted alcohol or carboxylic acid in the product obtained, the IR spectras were measured through Attenuated Total Reflectance (ATR) on a diamont tip. In particular, to validate the synthesis procedure, the spectra of commercial and synthesized DiMeOHOx were first compared.

Figure 1A shows a comparison between the IR spectra of the precursor oxalic acid (green), the synthesized DiMeOHOx (blue), and commercial DiMeOHOx (black). As a result of the similarity of the spectra of commercial and synthesized DiMeOHOx, and of the absence of peaks denoting the presence of unreacted oxalic acid, the synthesis procedure could be considered successful for the diesters.

It is important to consider that because the two carboxylic groups in DiMeOHOx are directly linked to each other with no additional carbons in between, the spectrum of DiMeOHOx appears slightly different than the ones of other esters or diesters [18,19]. Figure 1B shows the main differences between DiMeOHOx and DiMeOHSe for comparison. Diesters with longer carbon chains as DiMeOHSe generally present higher peaks at 2920 and 2850 cm^−1^ due to the stronger C-H stretching in the aliphatic chains.

In all cases, upon successful conversion of the carboxylic group to an ester group, a progressive disappearance in the sharp peak of carboxylic acids at 1705 cm^−1^ can be observed in favor of the appearance of the sharp peak at 1750 cm^−1^, which is typical of the stretching of O–C=O in ester bonds [26]. Additionally, the formation of the ester bond is visible in the peak at 1100 cm^−1^ from the stretching of C–O–C bonds, which is absent in both the precursor alcohol and acid. In case some unreacted acid remains in the product, both peaks from the stretching of HO–C=O from the acid and O–C=O from the ester are visible in the form of a split double-peak at 1705 and 1750 cm^−1^, respectively. The presence of traces of acid or alcohol can be also seen in the region between 2000 and 3000 cm^−1^. In particular, if unreacted alcohol is present in the product, a broad peak between 3600–3100 cm^−1^ arises, whereas the O–H stretching from carboxylic acids appears in the form of a broad peak between 3300 and 2100 cm^−1^.

Generally the IR spectra of different diesters of comparable purity appear almost unchanged, except for the difference in ratio between the sharp peaks of C–H stretching in the aliphatic chains at 2920 and 2850 cm^−1^ and the ester peak at 1750 cm^−1^. The longer the aliphatic chains, the higher the intensity of the peak at 2920 and 2850 cm^−1^ in comparison to the one at 1750 cm^−1^ [26].

#### 2.1.2. GC-MS

Although the ATR-IR could confirm the correct formation of the ester and exclude the presence of non-volatile impurities on a qualitative level, the possibility that other compounds were formed could not be dismissed. For example, if the monoester was to be formed, no discerning differences in the peak at 1750 cm^−1^ would be visible. Therefore, the compounds were analyzed through GC-MS to further confirm their purities and validate the hypothesis that only the diester is present in the bulk product.

Table 1 reports the retention times of the diesters in the GC column and the relative fragmentation peaks reported from the MS. Generally, only one peak shows in the chromatogram of all diesters; therefore, the presence of monoesters or other volatile compounds that would not be visible in the IR could be effectively rejected.

In the case of impurities from residual 1-pentanol or 1-decanol, a side peak at 3.29 min or 14.26 min respectively is visible. However, in all cases, such peaks are either absent or lower than 5% in intensity; thus, the purities of the diesters are assumed to be ≥95%. Additionally, the MS spectra and fragmentation patterns match, in all cases, with those in the official National Institute of Standards and Technology (NIST) database, which consents to confirm the identity of the diesters synthesized.

As can be observed in Table 1, the molecular ion is not visible in any of the compounds MS spectra, with the sole exception of DiMeOHOx at 118 mass-to-charge ratio (*m*/*z*). If present, the *m*/*z* of the molecular ion’s signal matches the value of the molecular mass of the diester.

Some similarities between the MS spectra of diesters derived from either the same alcohol or the same acid can be observed. In all pentyl diesters, the fragmented carbon chain from the alcohol sides is visible as a peak at 70 *m*/*z* with relative abundances varying from 10% as for DiPeOHSu to a maximum of 59% as in the case of DiPeOHSub. Similarly, in the case of decyl esters, the fragment corresponding to the loss of the alkyl carbon chains and the formation of a dicarboxylate ion is visible at 140 *m*/*z*, although with lower intensities. In methyl diesters, the progressive loss of one or two methanol molecules generates peaks of high intensities. For example, in the spectrum of DiMeOHSu, the loss of 31 *m*/*z* (methanol) from the molecular peak creates the most abundant peak at 115 *m*/*z*. A progressive loss of two molecules of methanol is hypothesized in DiMeOHSub and DiMeOHSe from the molecular ion at 171, 129 *m*/*z,* and 199, 166 *m*/*z* respectively, with intensities above 50% for the loss of the first alcohol molecule.

DiMeOHOx has been proven to undergo a different fragmentation path, which explains the lack of peaks corresponding to the loss of methanol. As reported by Liehr [27], the fragment at 59 *m*/*z* is the product of the C–C cleavage between the two carboxylic groups, whereas the one at 45 *m*/*z* corresponds to methoxymethyl ions formed via rearrangements from the molecular ion.

Some of the most abundant fragments in pentyl and decyl diesters, especially the longer chained ones, arise from the supposed formation of cyclic anhydrides from the loss of the alcohol chains and consequent cyclization [28]. The fragment corresponding to the formation of sebacic anhydride is visible in DiPeOHSe and DiDeOHSe in highest abundance at 185 *m*/*z*. The suberic anhydride appears at 157 *m*/*z* with intensities of 71% and 49% for DiPeOHSub and DiDeOHSub, respectively. Figure 2 shows the possible fragmentation path that induces the formation of anhydrides.

### 2.2. Thermal Characterization

After the structures and purities of the diesters were successfully confirmed, their thermal properties were characterized through DSC and TGA in terms of phase change transitions, enthalpies of fusion, and thermal degradation. Additionally, these properties were set against the carbon number to investigate the possible existence of trends correlating the chemical structure to the thermal behavior. Table 2 and Table 3 report all the average onset melting and crystallization temperatures measured for all diesters’ repetitions and the corresponding average enthalpies of fusion. The melting temperature of DiPeOHSu could not be defined, as the compound did not crystallize until the lower limit of the available DSC of −60 °C. Therefore, the corresponding missing values are reported in the tables as N.A. (Not Available).

The diesters present melting temperatures ranging from a minimum of −20 °C for DiPeOHSub to a maximum of 46 °C for DiMeOHOx. Although diesters are characterized by similar chemical structures as those of fatty esters, their thermal properties appear to be significantly different. Overall, diesters seem to be marked by lower melting temperatures compared to fatty esters of equal or similar chain length. For example, methyl palmitate has carbon number 17 and an onset melting temperature of approximately 26 °C [19], while DiPeOHSub has carbon number 18 but presents an onset melting point of −20 °C. This means that a single additional carbon in the chemical structure generates a difference in phase change temperature of more than 40 °C. Despite having the same number of 30 carbons in their aliphatic chains, decyl arachidate (DEAR) [19] and DiDeOHSe show onset melting at 41 °C and 31 °C, respectively. Therefore, diesters’ melting points seem to be consistently lower than those of comparable fatty esters.

The diesters also show a higher degree of supercooling compared to fatty esters, where supercooling was generally below 10 °C for samples in the milligrams range [19]. In the case of diesters, supercooling of up to 20 °C (DiPeOHOx) was observed.

The possible explanation offered by the authors relies on the assumption that ester groups are subjected to a certain degree of rotational freedom, as previously theorized by Bunn [29] and Crowe [30]. Thus, the presence of two ester bonds in diesters increases the overall rotational freedom of the molecule. Then, this would generate possible disruptions and vibrations in the crystal lattice, producing lower melting points and more fragile crystalline structures. The presence of the second ester bond could also cause bending and tangling, which lowers the final melting point of the compound and increases the tendency to supercool.

Figure 3 shows the onset melting points of each diester colored according to the corresponding alcohol of origin and plotted against the carbon number (Carbon Nr). Low deviations from the average onset melting temperature (*T_m_*) and crystallization temperature (*T_c_*) are observed for all compounds after six consecutive heating–cooling cycles, which hints at the thermal stability of the samples. Nevertheless, no conclusions can be drawn on their long-term stability, as further tests with additional cycles or longer times would be required.

As with that reported by Ravotti et al. [19] for fatty esters, pentyl diesters present consistently lower melting temperatures compared to diesters derived from methanol or decanol. This might be the result of an odd–even effect, where the odd-numbered 1-Pentanol forms molecules with lower symmetry and lower *T_m_*.

While the decyl diesters’ melting temperatures remain similar for growing carbon numbers, there is no clear behavior correlating the *T_m_* of pentyl and methyl diesters to their carbon numbers.

Besides DiMeOHOx, all decyl diesters have higher carbon numbers than the corresponding methyl and pentyl diesters derived from the same acid.

Interestingly, despite having the lowest carbon number of 4 amongst all diesters presented hereby and of all fatty esters presented by Ravotti et al. [19], DiMeOHOx is characterized by the highest melting point of the whole series (46.0 °C), comparable to that of methyl behenate [19] (47.9 °C) with carbon number 22. This could be explained with DiMeOHOx’s reduced bending and rotation, due to its short aliphatic chain, which produces higher T_m_ and more fixed geometries in the crystal lattice.

Figure 4 shows the enthalpies of fusion (Δ*H*) of each diester colored according to the corresponding alcohol of origin and plotted against the carbon number.

Similarly to what is observed for the melting points, in comparison with fatty esters, diesters present lower values with a minimum of 92 J/g for DiPeOHOx and a maximum of 172 J/g for DiDeOHSub. In particular, DiDeOHSe (C_30_) is the diester with the highest carbon number of the whole series, but it reports mild enthalpies of fusion around 160 J/g.

If DiDeOHSe (C_30_) is compared with the diester of tetradecanol with sebacic acid (DiTeOHSe, C_38_) produced by Aydin et al. [24], with enthalpies of approximately 202 ± 2 J/g, a difference of 8 carbon units produces a change of almost 45 J/g. Upon comparison between DiDeOHSe (C_30_) and DiDeOHOx (C_22_), the same difference of 8 carbon units produces a change of only 8 J/g in the resulting enthalpies. The reasons behind such variations are still unclear, but as pointed out by Malkin [31] for triglycerides, ΔH generally tends to increase for increasing carbon number. This is especially visible in Table 3. Here, the molar heat of fusions (Δ*H* J/mol) can be seen increasing for the increasing carbon number, for methyl and decyl esters, but not for pentyl esters. This seems to suggest a possible odd–even effect.

Methyl diesters and decyl diesters derived from the same acid show Δ*H* in a similar range of values, whereas pentyl diesters report consistently lower values.

Figure 5 shows the onset (blue) and endset (red) degradation temperatures (*T_degradation_*) of the diesters investigated. The diesters are ordered for increasing carbon number.

As expected, the stability of the diesters increases for the increasing carbon number, with the exception of DiMeOHSu, which shows lower onset and endset *T_degradation_*. DiMeOHOx and DiMeOHSu undergo mass losses ≥ 5% at temperatures as low as 75 and 85 °C due to their low carbon number and higher volatility, while DiDeOHSe show a 5% mass loss at 265 °C. All diesters were fully degraded up to 400 °C, as was the case with fatty monoesters [18,19]. No clear odd–even effects are observed.

## 3. Materials and Methods

All dicarboxylic acids (oxalic, succinic, suberic, and sebacic), methanol, 1-decanol, and tetradecanol were acquired from Sigma Aldrich (St. Louis, MO, USA) with high purities (≥98%) as synthesis precursors. Dimethyl oxalate was purchased from Sigma Aldrich with purity ≥ 99% and was used for comparison with the self-synthesized one for validation of the synthesis procedure. 1-Pentanol was purchased by Roth GmbH (Karlsruhe, Germany) with purity ≥ 98%. The synthesis procedure was the same as reported by Ravotti et al. [18]; thus, concentrated sulfuric acid (H_2_SO_4_) was used as the acid catalyst (Sigma Aldrich, ≥99%), sodium sulfate anhydrous (Na_2_SO_4_) was used as a water-absorbing agent (Sigma Aldrich, ≥99%) for the elimination of water, and ethyl acetate (EtOAc) was used as the organic solvent for extraction (Sigma Aldrich, ≥99%). Cylohexane (Sigma Aldrich, GC quality, ≥99.9%) was used as the solvent for the Gas Chromatography coupled with Mass Spectrometry (GC-MS) analysis. All chemical materials listed above were used without any prior purification.

### 3.1. Synthesis

All diesters presented in this study were synthesized from the corresponding carboxylic acids and alcohols and purified according to the method described by Ravotti et al. [18,19] and based on Fischer esterification with H_2_SO_4_ as a catalyst. Since dicarboxylic acids present two carboxylic groups, the excess of alcohol ratio used in comparison to the acid was 10:1. In contrast to this, the ratio proposed for fatty esters was 5:1. Each diester was synthesized three times in order to grant the reproducibility of the synthesis procedure. As the reaction mechanism was described in detail by Ravotti et al. [18,19], it will not be discussed further herein. To ensure purities ≥ 95%, the diesters were dissolved in methanol overnight with a ratio of approximately 10:1 alcohol:ester and crystallized at either −20 °C or +2 °C according to their range of melting temperatures. This procedure was repeated until no residual unreacted alcohol or acid could be observed through ATR-IR and GC-MS. Generally, up to three crystallizations for even the most impure diesters were sufficient to achieve the desired purity.

To further validate the synthesis, ditetradecyl sebacate (DiTeOHSe) from Aydin et al. [24] was reproduced three times with the synthesis procedure proposed in this study. The chemical structure and IUPAC name, IR spectra, DSC curves, and TGA curves are reported in Appendix B, Figure A2, Figure A3, Figure A4 and Figure A5. In the case of DiTeOHSe, the synthesis procedure was slightly modified. A ratio of 1.5:1 alcohol:ester was used, and sodium carbonate (Na_2_CO_3_) was added to the water phase during each washing with deionized water and EtOAc to precipitate any residual unreacted acid.

### 3.2. Characterization

#### 3.2.1. Differential Scanning Calorimetry (DSC)

The melting and crystallization temperatures and enthalpies of fusion ΔH were measured with a DSC 823^e^ and a DSC 3+ by METTLER TOLEDO (Columbus, OH, USA), which were both equipped with robotic autosamplers. The DSCs were calibrated with indium standards before the measurements, and the uncertainty of the instruments was ±0.1 K. A screening of the diesters’ phase change temperatures was first conducted with a general heating–cooling method between −60 and 100 °C with a heating rate of 10 K/min. Afterwards, each diester was cycled first three times at 10 K/min, and then three times at 2K/min in a temperature range around the melting temperature (*T_m_*) from *T_m_* − 30 °C to *T_m_* + 20 °C. Therefore, as the syntheses were repeated three times, each diester type was measured three times with six cycles per sample. The measurements were conducted under a constant flow of nitrogen at a rate of 100 mL/min, and sample masses were typically between 3 and 10 mg. The melting and crystallization peaks were calculated by the STAR^e^ software version 16.20 through the tangent method, and the enthalpies were obtained by integration of the corresponding peaks. The degree of supercooling was estimated using the difference between the onset melting peak and the onset crystallization peak reported by the instrument. The values reported were calculated as follows: for each diester, the mean for all six cycles at 10 K/min and 2 K/min for *T_m_*, *T_c_* and ΔH was derived; then, the mean of the three final values obtained for each repetition was retrieved and is reported alongside the relative standard deviation.

#### 3.2.2. Thermal Gravimetric Analysis (TGA)

The samples’ degradation temperatures were measured on a TGA STAR^e^ 2 System and a TGA/DSC 3+ by METTLER TOLEDO equipped with autosamplers in the temperature range between 25 and 600 °C with a heating rate of 10 K/min and sample masses between 15 and 20 mg. The uncertainty of the instrument’s balance is reported to be ±0.1 μg. For each sample list and method, a blank measurement of the empty crucible was performed with the same method as described above for baseline subtraction. The starting degradation temperature was typically defined as the earliest temperature for which mass losses ≥ 5% were observed. The end degradation temperature was defined as the earliest temperature for which the mass loss was ≥99%, and for which the mass loss curve in a plot of mass (mg) versus temperature (°C) was flat. The software used to analyze the data was the STAR^e^ software version 16.20.

#### 3.2.3. Attenuated Total Reflectance Infrared Spectroscopy (ATR-IR)

In order to confirm the presence of the ester bond and the absence of unreacted reagents, the structures were evaluated through Infrared Spectroscopy (IR) with an ATR-IR Cary 630 by Agilent Technologies (Santa Clara, CA, USA). The measurements were conducted in absorbance mode in the wavenumber range between 4000 and 600 cm^−1^ with 4 cm^−1^ resolution. No prior sample preparation was needed, and a background spectrum was registered every 10 min with 32 scans. Following the background collection, a few milligrams of the samples were deposited on the diamond’s surface in either solid or liquid form and the spectrum was registered with 32 scans.

#### 3.2.4. Gas Chromatography Coupled with Mass Spectrometry (GC-MS)

Gas chromatograms were measured on a Perkin Elmer (Waltham, MA, USA) GC Clarus 590 connected to a Perkin Elmer MS Clarus SQ 8 S with an EI standard source, with a 4.0 mm Glass inlet liner with deactivated wool in split mode 1:50 on a Perkin Elmer Elite 5MS 30 m × 0.25 mm × 0.25 μm capillary column and a 1.0 mL/min flow rate with injection volume of 1 μL. The oven was programmed as follows: 100 °C for 2 min, then 10 °C/min heating rate to 300 °C, and afterwards hold for 5 min. Mass spectra were measured with electron ionization (EI) at 70 eV, transfer line at 250 °C, and a source temperature of 200 °C. The mass fragments were scanned between 50 and 500 *m*/*z* for all diesters with the exception of dimethyl oxalate, where fragments were scanned between 40 and 500 *m*/*z*. In all cases, a scan time of 0.3 s and inter-scan delay of 0.04 s were used. The Mass Spectrometer (MS) was calibrated and tuned with perfluoroterbutylamine (PFTBA, Sigma Aldrich). The samples were diluted in cyclohexane with a concentration of 0.1 mg/mL.

## 4. Conclusions and Outlook

In this study, 12 diesters derived from four different dicarboxylic acids and three alcohols have been synthesized to high purities, and their thermal properties have been evaluated in terms of suitability for thermal heat storages applications. All syntheses proved to be successful and all chemical structures were confirmed via ATR-IR and GC-MS. To summarize, despite having a similar chemical structure to that of fatty esters, the diesters present different properties. The melting temperatures of the diesters with carbon numbers from 4 to 30 range from low temperatures of −20 °C DiPeOHSub to mid temperatures of 46 °C for DiMeOHOx. The onset melting points are lower compared to those of fatty esters of comparable chain length, with most pentyl diesters undergoing phase changes below 0 °C. This is supposed to be caused by an increase in the rotational freedom and the tendency to tangle due to the presence of a second carboxylic group in the chemical structure. This would also explain diesters’ higher tendency to supercool.

An odd–even effect is observed, according to which all diesters from 1-pentanol possess lower melting points than the corresponding methyl or decyl diesters derived from the same acid. This is probably caused by a resulting lower symmetry in the crystal lattice. The diesters from 1-decanol possess relatively similar melting points, whereas no clear trend can be identified correlating the melting points of methyl and pentyl diesters to their carbon numbers. This is in contrast to what Ravotti et al. [18,19] observed for fatty ester, and to what Malkin [31] and Lutton [32] observed for triglycerides. As such, it might be challenging to predict the diesters’ thermal properties [33,34], which might constitute a disadvantage in the future development of these PCMs. The enthalpies of fusion are also lower than those of fatty esters, with minimums of 92 J/g for DiPeOHOx to a maximum of 172 J/g for DiDeOHSe, whereas linear saturated fatty esters show enthalpies up to 200 J/g. In accordance to that observed by Malkin [31] for triglycerides, a trend can be seen correlating the carbon number to the molar heat of fusion (KJ/mol), according to which the enthalpies increase by increasing carbon number.

The values of the enthalpies and phase change transitions were stable for six consecutive heating–cooling cycles, which suggests that the diesters are thermally stable within the operational range of temperatures. However, further cycling tests are needed to confirm their long-term stability.

As with fatty esters, diesters show growing *T_degradation_* for growing carbon number, with the exception of DiMeOHSu, and a complete mass loss up to 400 °C for all samples. Although the enthalpies of fusion possessed by the diesters are quite mild for latent heat storage applications, some interesting candidates could still be identified. DiMeOHSub with its onset *T_m_* at −16 °C and a Δ*H* of 112 J/g could offer a sustainable solution for refrigeration processes in a temperature range where only few organic PCMs are found [35,36]. However, in the low to mid-temperature application range between 10 and 50 °C, where most commercial fatty esters show phase change transitions, the low values of Δ*H* and higher degrees of supercooling make diesters unfavorable candidates as organic PCMs.

In conclusion, fatty diesters could prove to be interesting organic substitutes to salt hydrates for refrigeration and cold processes in the subzero temperature range; however, for low to mid-temperature applications, their properties cannot compete with those of commercial fatty esters or paraffins.

## Figures and Tables

**Figure 1 materials-13-00810-f001:**
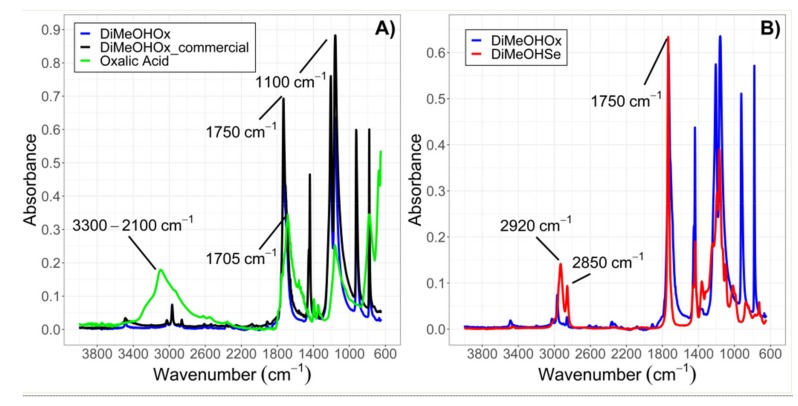
(**A**) Attenuated Total Reflectance (ATR)-IR spectra of commercial DiMeOHOx (black) and synthesized DiMeOHOx (blue) for comparison and synthesis validation. (**B**) Comparison between IR spectra of DiMeOHOx and DiMeOHSe. Due to the presence of longer carbon chains, DiMeOHSe presents enhanced peaks at 2820 cm^−1^ and 2920 cm^−1^.

**Figure 2 materials-13-00810-f002:**
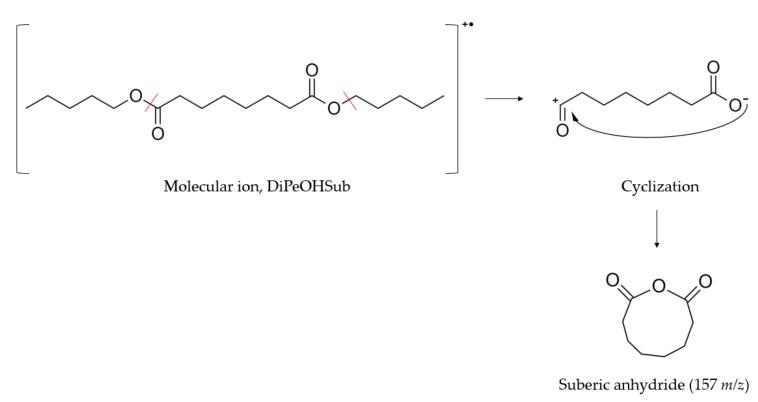
Proposed fragmentation and consecutive cyclization of diesters derived from 1-pentanol and 1-decanol. A cleavage of the alcohol on one side and the alkyl carbon chain on the other side occurs, followed by a closed ring formation. In the example, the mechanism for the formation of suberic anhydride from DiPeOHSub is reported, the peak of which appears at 157 *m*/*z* with 71% relative abundance.

**Figure 3 materials-13-00810-f003:**
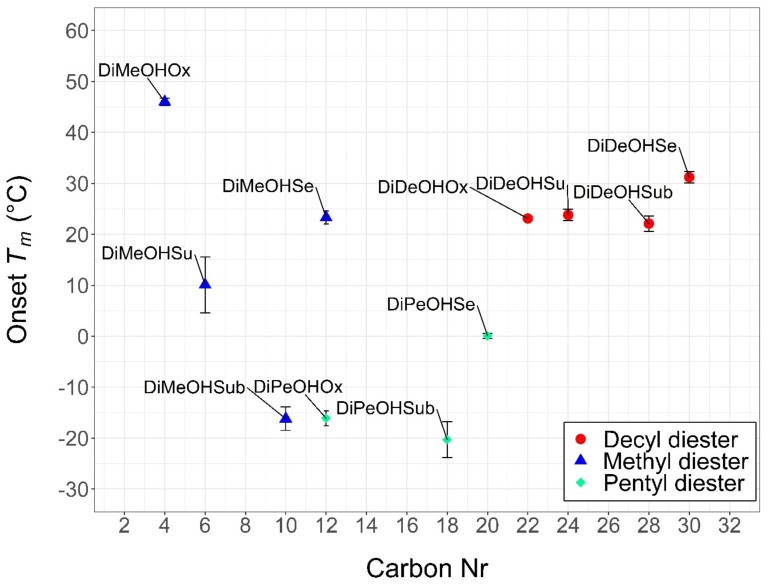
Scatter plot of the average onset melting points plotted against the carbon numbers of all the diesters reported in this study. The diesters have been grouped based on the corresponding precursor alcohol. Methyl diesters are indicated in blue, pentyl diesters are indicated in green, and decyl diesters are indicated in red. The error bars represent the standard deviation between the melting point of three repeated syntheses for each compound.

**Figure 4 materials-13-00810-f004:**
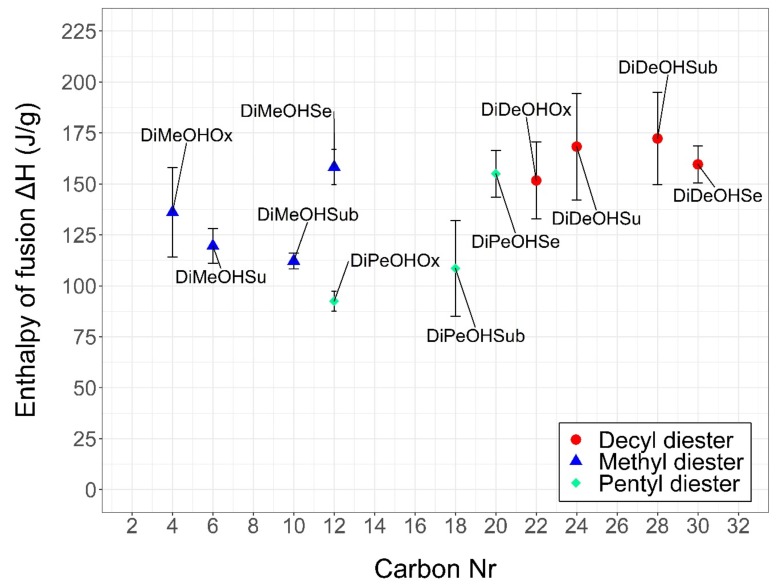
Scatter plot of the average enthalpies of fusion ΔH plotted against the carbon numbers of all diesters reported in this study. The diesters have been grouped based on the corresponding precursor alcohol. Methyl diesters are indicated in blue, pentyl diesters are indicated in green, and decyl diesters are indicated in red. The error bars represent the standard deviation between the means of three replicas for each compound.

**Figure 5 materials-13-00810-f005:**
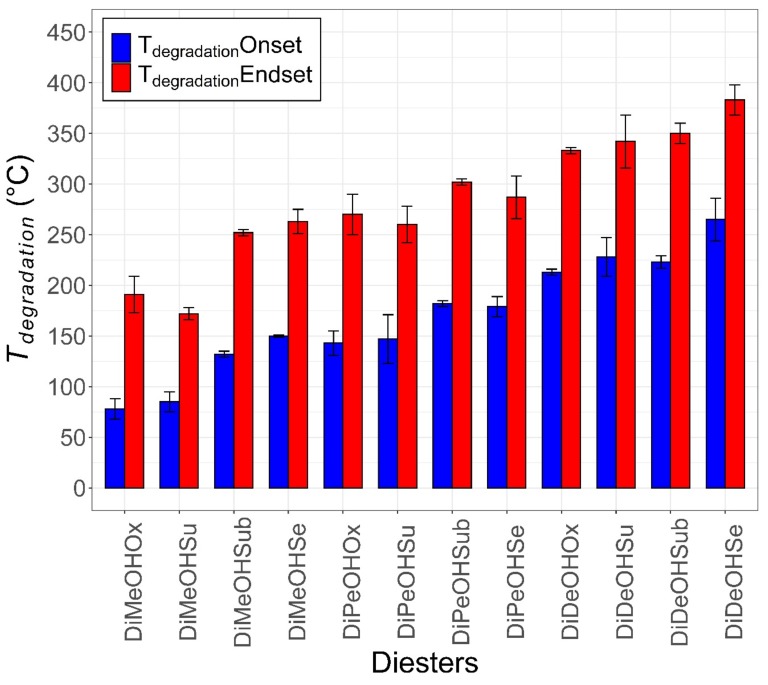
Bar plot showing the start (blue bar) and end (red bar) *T_degradation_* of the diesters presented. The onset T_degradation_ is defined as the earliest temperature at which the diesters undergoes mass losses ≥ 5%, while the endset T_degradation_ is the earliest temperature at which the mass loss ≥ 99%. The diesters are ordered according to increasing carbon number. A trend correlating the degradation temperatures to the carbon number can be seen, with the diesters with lower carbon numbers undergoing mass losses at lower temperatures.

**Table 1 materials-13-00810-t001:** Data obtained from the gas chromatograms and mass spectra of the diesters through GC-MS analysis.

Compound *	Retention Time GC, min	Molecular Weight (MW), g/mol	Fragmentation Peaks MS, *m*/*z* with Relative Intensities (%)
DiMeOHOx (1.2.1)	1.84	118	118 (8), 59 (100), 45 (32)
DiPeOHOx (5.2.5)	9.17	230	161 (1), 117 (5), 115 (8), 99 (1), 87 (3), 71 (100), 70 (55), 55(54)
DiDeOHOx (10.2.10)	18.27	370	281 (1), 207 (4), 191 (3), 169 (1), 164 (3), 140 (2), 133 (3), 124 (2), 112 (1), 97 (4), 85 (8), 83 (9), 71 (14), 69 (17), 57 (85), 55 (100)
DiMeOHSu (1.4.1)	2.73	146	116 (8), 115 (100), 114 (32), 101 (2), 87 (20), 86 (2), 60 (1), 59 (5), 57 (5), 56 (38), 55 (43)
DiPeOHSu (5.4.5)	12.12	258	189 (7), 171 (14), 144 (1), 119 (10), 102 (5), 101 (100), 74 (4), 71 (13), 70 (10), 69 (3), 55 (10)
DiDeOHSu (10.4.10)	19.71	398	259 (24), 241 (4), 207 (2), 141 (17), 119 (62), 111 (2), 101 (100), 97 (5), 85 (35), 83 (15), 71 (36), 69 (27), 57 (50), 55 (38)
DiMeOHSub (1.8.1)	7.94	202	171 (54), 139 (19), 138 (88), 129 (100), 114 (10), 110 (22), 101 (6), 97 (78), 87 (43), 83 (43), 74 (96), 69 (95), 67 (15), 59 (64), 55 (98)
DiPeOHSub (5.8.5)	15.30	326	227 (64), 200 (2), 185 (12), 157 (71), 139 (17), 138 (28), 129 (3), 115 (48), 111 (26), 97 (15), 83 (27), 70 (59), 55 (100)
DiDeOHSub (10.8.10)	22.43	454	327 (1), 297 (27), 281 (1), 253 (2), 209 (1), 157 (49), 140 (10), 115 (16), 111 (15), 97 (20), 83 (32), 70 (43), 69 (65), 57 (58), 56 (67), 55 (100)
DiMeOHSe (1.10.1)	10.30	230	200 (3), 199 (54), 170 (4), 166 (42), 157 (45), 148 (6), 139 (20), 138 (53), 125 (98), 121 (18), 111 (13), 107 (7), 101 (9), 98 (72), 97 (57), 87 (36), 84 (54), 83 (49), 79 (8), 74 (96), 73 (12), 69 (42), 67 (16), 59 (42), 55 (100)
DiPeOHSe (5.10.5)	17.00	342	256 (7), 255 (72), 228 (2), 213 (15), 186 (10), 185 (100), 166 (12), 143 (32), 139 (18), 125 (26), 121 (10), 98 (25), 97 (21), 83 (12), 73 (7), 70 (38), 69 (20), 60 (4), 55 (39)
DiDeOHSe (10.10.10)	23.88	483	325 (50), 283 (4), 253 (2), 207 (5), 186 (6), 185 (100), 166 (4), 143 (10), 139 (8), 125 (10), 111 (4), 97 (18), 83 (18), 70 (29), 69 (38), 57 (43), 55 (50)

* In the “compound” column, the abbreviation of the diesters’ names is indicated alongside the carbon number of the alcohols and the acid in brackets. The carbon number of the corresponding precursor acid and alcohol are reported in brackets. For example, DiMeOHOx is the product of two methyl groups (C_1_) attached at the extremes of a central oxalic acid molecule (C_2_). Therefore, it is indicated as 1.2.1. Next, the retention times taken by the compounds to exit the GC column are reported in minutes and the molecular weight of each diester is expressed in grams per mole. In the last column on the right, the main *m*/*z* fragments registered by the MS are noted alongside their relative intensity percentages within brackets. If present, the molecular ion is the first peak that appears in the fragmentation column, with *m*/*z*’s value matching that of the molecular weight. It is visible for DiMeOHOx at 118 *m*/*z* with an intensity of 8%.

**Table 2 materials-13-00810-t002:** Summary of thermal properties of the diesters produced.

	Structure	Carbon Number	MW (g/mol)	*T_c_* * (Onset, °C)	*T_m_* * (Onset, °C)	Supercooling (°C)	Δ*H* * (J/g)	Δ*H* (KJ/mol)	*T_degradation_* * (Start, °C)	*T_degradation_* * (End, °C)
DiMeOHOx	C_4_H_6_O_4_	4	118	30.2 ± 2.3	46.0 ± 0.7	15.7	136.1 ± 22.0	16.1	78 ± 10	191 ± 18
DiDeOHSe	C_30_H_58_O_4_	30	483	30.1 ± 1.1	31.2 ± 1.1	1.2	159.6 ± 9.1	77.1	265 ± 21	383 ± 15
DiDeOHSu	C_24_H_46_O_4_	24	398	17.5 ± 1.2	23.8 ± 1.1	6.3	168.3 ± 26.2	67.0	228 ± 19	342 ± 26
DiMeOHSe	C_12_H_22_O_4_	12	230	5.3 ± 3.0	23.3 ± 1.3	18.0	158.3 ± 8.6	36.4	150 ± 1	263 ± 12
DiDeOHOx	C_22_H_42_O_4_	22	370	20.5 ± 0.7	23.1 ± 0.3	2.6	151.7 ± 18.8	56.1	213 ± 3	333 ± 3
DiDeOHSub	C_28_H_54_O_4_	28	454	18.2 ± 1.6	22.1 ± 1.5	3.9	172.3 ± 22.7	78.2	223 ± 6	350 ± 10
DiMeOHSu	C_6_H_10_O_4_	6	146	−6.8 ± 6.6	10.1 ± 5.5	16.9	119.6 ± 8.6	17.5	85 ± 10	172 ± 6
DiPeOHSe	C_20_H_38_O_4_	20	342	−3.3 ± 0.2	0.1 ± 0.5	3.4	155.0 ± 11.4	53.0	179 ± 10	287 ± 21
DiPeOHOx	C_12_H_22_O_4_	12	230	−36.3 ± 3.0	−16.1 ± 1.5	20.2	92.5 ± 4.9	21.3	143 ± 12	270 ± 20
DiMeOHSub	C_10_H_18_O_4_	10	202	−26.0 ± 2.6	−16.2 ± 2.3	9.8	112.2 ± 3.8	22.7	132 ± 3	252 ± 3
DiPeOHSub	C_18_H_34_O_4_	18	326	−28.4 ± 2.3	−20.3 ± 3.5	8.1	108.6 ± 23.4	35.4	182 ± 3	302 ± 3
DiPeOHSu	C_14_H_26_O_4_	14	258	N.A.	N.A.	N.A.	N.A.	N.A.	147 ± 24	260 ± 18

* *T_m_* stands for melting temperature, *T_c_* stands for crystallization temperature, Δ*H* stands for enthalpy of fusion, and *T_degradation_* stands for the start and end for the onset and endset degradation temperatures, respectively. The diesters are ordered according to decreasing onset *T_m_*. The colors indicate the temperature gradient from the highest values (red) to the lowest ones (blue). The onset *T_m_* and *T_c_* values are the result of the mean between the measurements performed on each of the three synthesis repetitions for each compound type. The Δ*H* values reported indicate the mean of the values registered for six consecutive heating–cooling cycles at 10 K/min and 2 K/min. All uncertainties represent the standard deviations between three replicated compounds. The supercooling is calculated as the difference between the average onset *T_m_* and the average onset *T_c_*.

**Table 3 materials-13-00810-t003:** Data shown in Table 2, reordered according to decreasing *T_degradation_* end, from the highest values (red) to the lowest ones (blue).

	Structure	Carbon Number	MW (g/mol)	*T_c_* (Onset, °C)	*T_m_* (Onset, °C)	Supercooling (°C)	Δ*H* (J/g)	Δ*H* (KJ/mol)	*T_degradation_* (Start, °C)	*T_degradation_* (End, °C)
DiDeOHSe	C_30_H_58_O_4_	30	483	30.1 ± 1.1	31.2 ± 1.1	1.2	159.6 ± 9.1	77.1	265 ± 21	383 ± 15
DiDeOHSu	C_24_H_46_O_4_	24	398	17.5 ± 1.2	23.8 ± 1.1	6.3	168.3 ± 26.2	67.0	228 ± 19	342 ± 26
DiDeOHSub	C_28_H_54_O_4_	28	454	18.2 ± 1.6	22.1 ± 1.5	3.9	172.3 ± 22.7	78.2	223 ± 6	350 ± 10
DiDeOHOx	C_22_H_42_O_4_	22	370	20.5 ± 0.7	23.1 ± 0.3	2.6	151.7 ± 18.8	56.1	213 ± 3	333 ± 3
DiPeOHSub	C_18_H_34_O_4_	18	326	−28.4 ± 2.3	−20.3 ± 3.5	8.1	108.6 ± 23.4	35.4	182 ± 3	302 ± 3
DiPeOHSe	C_20_H_38_O_4_	20	342	−3.3 ± 0.2	0.1 ± 0.5	3.4	155.0 ± 11.4	53.0	179 ± 10	287 ± 21
DiPeOHOx	C_12_H_22_O_4_	12	230	−36.3 ± 3.0	−16.1 ± 1.5	20.2	92.5 ± 4.9	21.3	143 ± 12	270 ± 20
DiMeOHSe	C_12_H_22_O_4_	12	230	5.3 ± 3.0	23.3 ± 1.3	18.0	158.3 ± 8.6	36.4	150 ± 1	263 ± 12
DiPeOHSu	C_14_H_26_O_4_	14	258	N.A.	N.A.	N.A.	N.A.	N.A.	147 ± 24	260 ± 18
DiMeOHSub	C_10_H_18_O_4_	10	202	−26.0 ± 2.6	−16.2 ± 2.3	9.8	112.2 ± 3.8	22.7	132 ± 3	252 ± 3
DiMeOHOx	C_4_H_6_O_4_	4	118	30.2 ± 2.3	46.0 ± 0.7	15.7	136.1 ± 22.0	16.1	78 ± 10	191 ± 18
DiMeOHSu	C_6_H_10_O_4_	6	146	−6.8 ± 6.6	10.1 ± 5.5	16.9	119.6 ± 8.6	17.5	85 ± 10	172 ± 6

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
