# Peer review of "Investigation of the Thermal Properties of Diesters from Methanol, 1-Pentanol, and 1-Decanol as Sustainable Phase Change Materials"

_materials, 2020, doi:10.3390/ma13040810_

Round 1

Reviewer 1 Report

interesting and innovative work,

carefully written in accordance with the editing requirements,

has scientific and application value,

in my opinion ready for publishing. 

Author Response

Dear Madam or Sir,      Thank you very much for your review. We are pleased to know you enjoyed the manuscript and find the results presented interesting.   We are very much glad to know you think the manuscript is ready for publishing. We kindly offer you our Best Regards.  

Reviewer 2 Report

The manuscript “Investigation of the Thermal Properties of Diesters from Methanol, 1-Pentanol and 1-Decanol as Sustainable Phase Change Materials” by Ravotti et al. describes the synthesis and physical analysis of twelve diesters regarding their thermal properties as phase change materials. The study extents the author’s previous studies on ester-based phase change materials and they applied Differential Scanning Calorimetry and Thermogravimetric Analysis to assess the thermal properties of the new diester materials. Although these diesters are less suited for low and mid temperature usage, the authors found a promising diester with advantageous thermal properties for potential cooling applications. The manuscript is clearly written and the experiments are technically sound. I have only a minor point: The newly synthesized diesters should be characterized using NMR spectroscopy, since NMR is the standard technique used to examine and confirm the successful synthesis of new entities.

Author Response

Dear Madam or Sir,

Thank you very much for your insightful comments and inputs. 

We are pleased to know the manuscript is of your interest and is technically sound in your opinion.

Regarding the NMR measurements, we agree that NMR is usually employed to verify new chemicals' synthesized structures.

However, based on our experience (please refer to the following articles: 1) Appl. Sci. 2018, 8(7), 1069; https://doi.org/10.3390/app8071069, and 2) Appl. Sci.2019, 9(2), 225; https://doi.org/10.3390/app9020225), we have usually encountered good agreement between the results obtained from ATR-IR, GC-MS and NMR.

Therefore, as we were able to confirm the structures (validated by NIST database as well) and the degrees of purities of the diesters presented hereby via ATR-IR and GC-MS, we deemed the NMR unnecessary for the present study.

We hope you can find this explanation satisfying.

Thank you and Best Regards. 

Reviewer 3 Report

In this work, “Investigation of the thermal properties of diesters from methanol, 1-pentanol and 1-decanol as sustainable phase change materials”, the authors assessed twelve diesters (from four dicarboxylic acids) and three alcohols as bio-based phase-change materials for thermal energy storage. This manuscript has a strong potential for a second review after applying the issues and addressing the shortcomings listed below:

1-Overall, the authors should polish/revise some grammatical mistakes and typos along the manuscript. For instance, ‘…up to date their thermal…’, ‘…phase change materials…’, ‘…scanning calorimetry and thermogravimetric analysis’, ‘Similarly to other…’, ‘…diamont tip’, ‘“show”, too much repetition along the manuscript. The authors should consider to use synonyms’. From now on, I am not mentioning the remaining ones here. I invite the authors to carefully read their work and make the required changes where necessary.   

2-Revise the following statements along the manuscript: ‘…amongst which are a better sustainability and a greater…’, ‘…remain for the most part…’, ‘The diesters investigated present…’. From now on, I am not specifying the remaining ones here. I invite the authors to carefully read their manuscript and conduct the required changes where necessary. 

3-What was the main reason behind assessing oxalic, succinic, suberic, and sebacic (as four dicarboxylic acids) and methanol, 1-pentanol, and 1-decanol (as three alcohols) as bio-based PCMs (why them, but not other sort of bio-based PCMs)?

4- I would request the authors to not use “ATR-IR, GC-MS, DSC, TGA” within the Abstract section. From my perspective, there is no need to mention them in the Abstract. Indeed, they have already mentioned them in the Introduction section.

5-The Introduction part is lack of the recent advancements in the field of phase-change materials and their applications. The following works must be properly mentioned and cited within the manuscript for the broad range of possible readers of the submitted manuscript: [(i) Adv. Optical Mater. 7, 1900171 (2019); (ii) Photonics 6(2), 52 (2019); (iii) J. Appl. Phys. 120, 164504 (2016)].

6-In Fig. 1, update the caption for 1B. Besides, update the order of produced diesters (based on the list that you presented on the left side of Table 1) in Table 2 and 3, to make a valid comparison.

7-Make the “Conclusions and Outlook” section more compact for legibility. There is no need to start a new paragraph so frequently.

8-Update Fig. B3. In its current form, it is really hard to capture the legends within the figure (Do this also for Fig. B4). Likewise, increase the linewidth of each data.

Author Response

Dear Madam or Sir,

Thank you very much for your insightful comments and inputs. 

Please find below a detailed description of all modifications/additions performed on the manuscript following your requests:  

1)-2) Generally, the manuscript has been proofread and reviewed by a native speaker with relevant technical background. Some corrections have been apported to adjust the issues reported. 

3) Some lines have been added in the introduction to clarify the reasons behind the diesters' choice.

4) The abbreviations have been deleted from the abstract as requested.

5) The references mentioned have been added in the introduction as requested.

6) The reason behind the diesters' order in the tables is underlined in the respective captions. The diesters are ordered according to specific properties in order to allow the reader to identify more easily trends correlating the chemical structures to the thermal behaviour.

7) The conclusions and outlook section has been compacted as requested.

8) Figure B3 and B4 have been updated. More specifically, the font size and line width have been increased, and the hatching in the curves area has been eliminated to allow the reader to view the curves more easily.

We hope the changes suit your remarks.

Thank you and Best Regards. 

Reviewer 4 Report

This is an extremely well-prepared manuscript. The authors sythensized 12 diesters and studied their thermal properties as phase change materials for applications in thermal energy storage. The data and experimental results regarding these chemicals can serve as benchmark for future researchers in this field to reference and implement. One minor typo in Table 1 needs to be fixed, for DiMeOHOx, the m/z value for methoxymethly ions should be 45 instead of 49.

Thanks.

Author Response

Dear Madam or Sir,      Thank you very much for your review. We are pleased to know you enjoyed the manuscript and find the results presented interesting.   The typo mentioned has now been fixed in Table 1.   Thank you and Best Regards.  

Round 2

Reviewer 3 Report

In its current form, the revised manuscript is suitable for publication.